# Gesture Evaluation in Virtual Reality

### Axel Wiebe Werner
axelww@kth.se
KTH Royal Institute of Technology
Stockholm, Sweden

### Jonas Beskow
beskow@kth.se
KTH Royal Institute of Technology
Stockholm, Sweden

### Anna Deichler
deichler@kth.se
KTH Royal Institute of Technology
Stockholm, Sweden

## Abstract

Gestures play a crucial role in human communication, enhancing interpersonal interactions through non-verbal expression. Burgeoning technology allows virtual avatars to leverage communicative gestures to enhance their life-likeness and communication quality with AI-generated gestures. Traditionally, evaluations of AI-generated gestures have been confined to 2D settings. However, Virtual Reality (VR) offers an immersive alternative with the potential to affect the perception of virtual gestures.

This paper introduces a novel evaluation approach for computer-generated gestures, investigating the impact of a fully immersive environment compared to a traditional 2D setting. The goal is to find the differences, benefits, and drawbacks of the two alternatives. Furthermore, the study also aims to investigate three gesture generation algorithms submitted to the 2023 GENEA Challenge and evaluate their performance in the two virtual settings.

Experiments showed that the VR setting has an impact on the rating of generated gestures. Participants tended to rate gestures observed in VR slightly higher on average than in 2D. Furthermore, the results of the study showed that the generation models used for the study had a consistent ranking. However, the setting had a limited impact on the models' performance, having a bigger impact on the perception of 'true movement' which had higher ratings in VR than in 2D.

## CCS Concepts

• **Human-centered computing** → *Empirical studies in interaction design.*

## Keywords

virtual reality, gesture generation, embodied conversational agents, evaluation paradigms, dyadic interaction

**ACM Reference Format:**
Axel Wiebe Werner, Jonas Beskow, and Anna Deichler. 2024. Gesture Evaluation in Virtual Reality. In *INTERNATIONAL CONFERENCE ON MULTI-MODAL INTERACTION (ICMI Companion '24), November 4–8, 2024, San Jose, Costa Rica.* ACM, New York, NY, USA, 9 pages. https://doi.org/10.1145/3686215.3688821

## 1 Introduction

In the realm of human communication, gestures serve as an integral component, facilitating non-verbal expression and enhancing the richness of interpersonal interactions [21]. Gestures can convey many different types of information between speaker and interlocutor, ranging from clear communication of intent, *e.g.*, the thumbs up gesture [12], to ambiguous and non-codified gestures which may convey some subconscious thought [9] or emotional state [14].

A burgeoning technology that seeks to utilize the communicative quality of gesticulation is the field of Embodied Conversational Agents (ECAs) [19][5], in which the study of gestural communication gains prominence as researchers seek to produce AI-generated gestures to increase the life-likeness and communicative qualities of ECAs [15]. So far, the evaluations of these generated gestures have mainly been conducted in 2D settings. However, as technology continues to advance, VR offers a novel platform for communication that enables people to engage in realistic and immersive environments. The 3D and interactive nature of VR has the potential to revolutionize how gestures are perceived and interpreted compared to their portrayal in conventional 2D settings. This research aims to dissect the nuances of gestural communication by scrutinizing the impact of the environment on the perception and interpretation of gestures and to perform a comparative evaluation of gestures exchanged between a speaker and an interlocutor in a dyadic setting, focusing on the differences between the immersive experience of VR environments and the traditional 2D setting.

To achieve a comprehensive understanding, this study employs a comparative evaluation where the observer will evaluate avatars in a number of different scenarios both monadic (one avatar) and dyadic (two avatars), with a particular focus on the subtleties of gestural communication. By conducting a systematic evaluation, we seek to elucidate whether VR environments foster a more authentic and nuanced perception of gestures, thereby enhancing the overall communication experience compared to interactions in 2D settings.

The significance of this research lies not only in advancing our understanding of the gestural capabilities of ECAs in immersive environments but also in providing practical insight into the differences between immersive and non-immersive environments. As VR and AI-powered conversational agents become increasingly integrated into our daily lives, a nuanced evaluation of their combined impact on communication is productive. This work seeks to contribute valuable insights that may inform the development of more immersive and effective communication agents.

### 1.1 Research Questions

With this in mind, the questions we seek to answer are the following.

(1) Are there significant differences between the evaluation of computer-generated gestures in 2D vs in VR? If so, what pros and/or cons do the different mediums provide?
(2) How do the three gesture generation models perform in VR compared to 2D settings, and what are the differences or similarities in their effectiveness based on a ground-truth human movement system?

## 2 Related Work

Efficient and accurate methods to evaluate gesture generation systems are more relevant than ever given the steady stream of generative gesture models being developed today. In a recent article by Wolfert *et al.* [24], three methods for the evaluation of computer-generated nonverbal behavior were tested and compared. The goal was to compare the three evaluation methods in how well they could record subtle forms of nonverbal behavior such as listening behaviour, i.e. backchannel communication, in a dyadic setting.

The first method was a direct rating method for human-likeness of gesticulation. Three videos were available at the same time as per the HEMVIP framework [11], and the observer would rate each video a score from 0 (worst) to 100 (best), similar to the scenario used in the 2023 GENEA Challenge (see below).

The second method was a direct rating of gesture appropriateness in which matching and mismatching stimuli were presented to the observer simultaneously. What this means is that one video in which the agent acted on natural (motion-captured) data was placed next to a video in which the agent acted on generated data. The observer was then asked to identify which was the natural motion and which was generated. Here, the authors used Barnard's test [4] to identify statistically significant differences between conditions at the level of $\alpha = 0.05$ while additionally applying the Holm–Bonferroni [10] method to correct for multiple comparisons.

The third and final method was a questionnaire-based study. The subjects were presented with 8 videos with a questionnaire of 15 questions after each video.

The study found that direct rating methods were better suited for this type of evaluation, especially with regard to providing deeper insight into the more subtle non-verbal communication. The questionnaire was less sensitive than directly rating the quality of the motion, as it did not detect subtle qualitative differences in behavior and was not calibrated between the raters. The questionnaire also had lower inter-rater reliability, meaning that different raters gave inconsistent ratings for the same motion stimuli. Finally, the questionnaire took much longer to complete than direct rating methods, and the authors suspected that it might have led to tester fatigue, which in turn led to poorer results.

In a more comprehensive overview of gesture evaluation methods, the authors of a 2022 article review 22 studies that use different methods to create co-speech gestures for ECA, such as rule-based and data-driven approaches, as well as the different evaluation methods used in these studies [25] and found a number of interesting trends that are valuable to consider when designing an evaluation of co-speech gestures for ECA. The authors arrive at a set of guidelines regarding e.g. participant sample, test set-up, and measurement type.

The most comprehensive concerted effort on gesture evaluation is the GENEA Challenge [15]. This is an open and recurring challenge to evaluate speech-driven gesture generation systems for ECA that started in 2021. The 2023 challenge provided data on both sides of a dyadic interaction, allowing teams to generate full-body motion for an agent given its speech and the speech and motion of the interlocutor. The challenge evaluated 12 submissions and 2 baselines together with held-out motion-capture data in three large-scale user studies, focusing on 1) the human-likeness of the motion, 2) the appropriateness of the motion for the agent's own speech, and 3) the appropriateness of the motion for the behavior of the interlocutor in the interaction.

The HEMVIP methodology [11] was used to measure how human-like the motion of the virtual characters appeared, without considering the speech or the interlocutor's behavior. Participants rated the motion on a sliding scale from 0 (worst) to 100 (best) for each segment.

Appropriateness for agent speech was evaluated using a mismatching methodology to measure how well the motion matches the speech of the main agent, controlling for the human-likeness of the motion. Participants chose one of five options to indicate their preference between a pair of videos, where one video had matched motion and speech, and the other had mismatched motion and speech.

Finally, the evaluation of interlocutor behaviour appropriateness also used a mismatching methodology but focused on how well the motion matches the behaviour of the interlocutor (both speech and motion), controlling for the human-likeness of the motion and the agent's own speech. Participants chose one of five options to indicate their preference between a pair of videos, where one video matched motion and interlocutor behavior, and the other had mismatched motion and interlocutor behaviour.

After evaluation, various statistical tests were applied to the user ratings, such as the Mann-Whitney U test [20], Welch's t test [22], and the Holm-Bonferroni correction [10], to determine the significance of differences between conditions and to control for multiple comparisons and false discovery rate. The results showed a large span in human-likeness between challenge submissions, with a few systems rated close to human motion capture. Appropriateness seems far from being solved, with most submissions performing in a narrow range slightly above chance, far behind natural motion, meaning that ratings were close to random and not signifying of any trend. The effect of the interlocutor is even more subtle, with submitted systems at best performing barely above chance.

Common to all of the above evaluation efforts is their reliance on 2D environments, such as web browsers, which are standard for subjective assessments. However, virtual reality (VR) and extended reality (XR) have seen limited utilization in gesture evaluation, with few exceptions like [7]. Given the growing prevalence of VR/XR applications featuring virtual agents (e.g., [18], [16], [17], [8]), this work aims to investigate two key questions: the extent to which existing evaluation paradigms can be effectively transferred to VR, and the potential for VR-based settings to offer unique insights beyond traditional 2D environments for evaluating gestures in virtual agents.

## 3 Method

### 3.1 Data preparation

The experiment relied on data provided by the 2023 GENEA Challenge, which is accessible through an open access online library [3]. The library provides many types of data, of which this study used the following categories:

- ML-generated BVH-data for main agent gesticulation
- Motion captured BVH-data for main agent gesticulation
- Audio files of the main agents speech with related transcribed files
- Motion captured BVH-data for interlocutor gesticulation
- Audio files of the interlocutors speech with related transcribed files

BVH-data refers to the data used to animate the avatars. From these files, two sets of clips were extracted. The first set of clips was chosen based on the following criteria:

- Length between 7-15 seconds long.
- The main agent is the main speaker.
- The clip only contains full sentences.

This set contains 21 files and is used for the first test scenario in which no audio is used, only the motions of the avatars. The second set, which contains 38 clips, was chosen with the same criteria as the first set but with the addition that the audio files did not contain any anonymized data since such data was cut out of the audio. This set was used for the second and third test scenarios, where the gestures were accompanied by audio.

In order to find these clips, the text logs of the conversations were first manually scanned to find segments where the main agent was the main speaker. After processing all logs, the audio files corresponding to those segments were reviewed to determine the suitability of each segment. Once the final files were chosen, a Python script was used to cut both the BVH and the WAV files into the selected segments.

The final step in preparing the data was to bake the BVH files into an avatar. For this, Motionbuilder 2024 was used. The files were then exported as FBX files which could be imported directly into Unity.

### 3.2 Systems

The GENEA Challenge 2023 [15] included 14 systems plus ground truth motion (motion capture). In this study, we chose to include the three top-ranking systems from the GENEA 2023 human-likeness evaluation:

**SG** Diffusion-based system that uses contrastive pre-training of speech- and text embeddings to extract features to drive the diffusion model [6]

**SF** Diffusion-based system taking a variety of inputs including position, velocity, acceleration, rotation, pitch, and energy [26]

**SJ** Transformer-based system taking speech- and text embeddings and speaker identity labels as input [23]

In addition to data from the above models, we also included ground truth motion clips, **GT**.

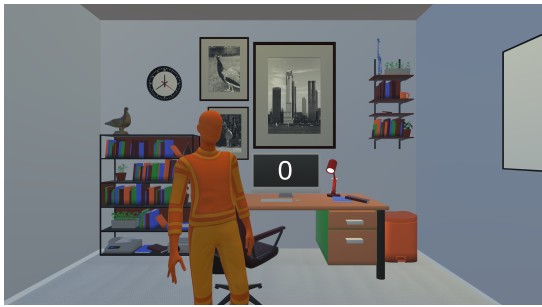

**(a) Monadic**

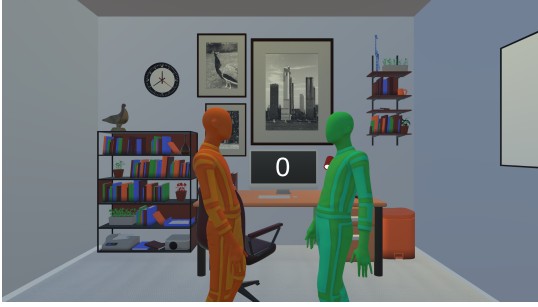

**(b) Dyadic**

**Figure 1: Environment used in experiment**

### 3.3 Pilot testing

Two pilot tests were run in order to fine-tune the experiment environment, information texts, and the test procedure. The pilot tests consisted of running parts of the test, both in 2D and in VR, with one subject and an extra observer in order to get feedback from "within" and "without" the test. Any feedback could then be synthesized into the project for improvement.

## 4 Environment design

The design of the environment in which the participants would observe the avatars was done in Unity using free resource packs from the Unity store. The aim was to design an environment that felt somewhat real in order to enhance immersion, without being too distracting from the goal of the experiment. This is different from most prior studies in which a very plain background was used, and the rationale for adding more detail in this experiment was to make the procedure more enjoyable to participants and therefore hopefully increase attention and decrease fatigue. Therefore, an office setting was chosen; see Figure 1

### 4.1 VR Implementation

The VR implementation was done using the OpenXR kit [13], which provides preset resources for VR programs. More specifically, it was the XR Interaction Toolkit sample that provided the pre-made resources mostly relied on for the VR scenario design. Scripting for the unity environment was done in C#. Six scripts were written in order to load in the necessary data (animation- and audio files), animate the avatars, play audio, and save the results to a CSV file.

The hardware used for the implementation was the Oculus Quest 2 VR headset with controllers.

## 4.2 Experiments

Before the experiment, informed consent was obtained from the subject. Participants were informed about the voluntary nature of their participation and how their data would be used and stored. In addition, participants were informed of any potential risks or discomfort associated with using VR technology, such as motion sickness or visual discomfort.

This was followed by a pre-test questionnaire in which information was submitted about gender, age, cultural belonging, and familiarity with VR, all data that could be used to check for potential bias. After this, the test was explained and the participant was familiarized with the tasks they were to perform.

The experiment itself was an observed user test which consisted of three scenarios that the subject would observe once in VR and once in 2D for a total of six tests.

**Scenario 1** The first scenario featured one avatar and included no audio. The participant was asked to focus on and rate only the naturalness of the avatars motion. As such this scenario was designed to measure the naturalness of the generated motions.

**Scenario 2** In the second scenario the participant observed a conversation between two avatars and was asked to focus on and rate how appropriate the main agent's gestures were to what the main agent was saying. This scenario therefore measured the speech appropriateness of the generated motions.

**Scenario 3** The third scenario had the same setup as the second but now the participant was asked to focus on how appropriate the main agent's gestures were to the interlocutor's gestures and speech, i.e. the conversational flow. This was meant to measure the dyadic appropriateness of the generated motions.

The participants watched 12 clips for each scenario and after each clip provided a direct rating. The clips were mixed in a random order to make sure that there was no intrinsic bias from clip order, and the order of the settings was switched between each participant *i.e.,* one participant started with 2D and then went to VR and the next participant started in VR and then went to 2D. While the interlocutors' movements were based on motion capture, the main agent's movements were based on four different systems: the ground truth (GT) and the three ML models previously mentioned (SG, SJ and SF). Three clips from each system were used for each scenario. All experiments were observed by a researcher who answered questions and made sure the process flowed smoothly, as well as ensuring the participants' attention on the tasks.

After completing the experiment, the participants were asked to fill out two post-test questionnaires, a modified version of the NASA TLX [2] and a version of the IGROUP Presence Questionnaire [1] to provide qualitative insights into their experiences and interpretations of the gestures and the test itself, as well as any feedback on the study, such as suggestions for improvement or concerns about the research process.

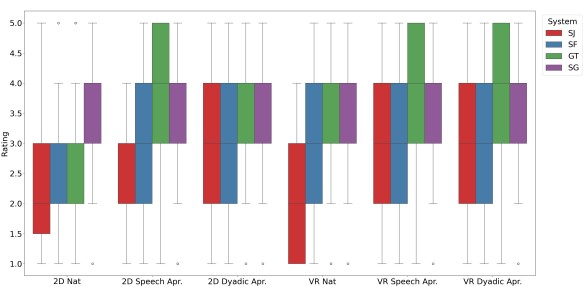

**Figure 2: Distribution of ratings per system across all scenarios**

Furthermore, no personally identifying data was collected throughout the experiment as a measure to reduce the risk of privacy breaches. Only parameters such as age, preferred gender, cultural belonging, *etc.* were collected, ensuring the anonymity of the participants.

In total, 30 participants were recruited for the test (age: min = 22, median = 25, max = 68; female: 9, male: 21), nearly all associating with a Northern European belonging. The tests generated a total of 2160 ratings, 1080 ratings per setting (2D and VR), 540 ratings per system, and 360 ratings per scenario. The results of the ratings showed some interesting differences between settings, systems and scenarios, the significance of which will be analyzed below.

## 5 Results

## 5.1 General results

In Figure 2 the distribution of ratings across system per scenario is displayed. From this, it is obvious that some systems performed better than others and that some scenarios were generally lower rated. More specifically we can see that the GT system, which is motion-captured data, performed at the top or at the shared top in all scenarios except the Naturalness scenario in the 2D setting. It is also evident that the SJ system performed at the bottom or the shared bottom in all scenarios. Furthermore, we can see that the performance of the SG model was consistent across all scenarios. The SF model had a large spread over almost all scenarios, steadily in the middle.

The statistical significance of these rating differences was investigated using a Wilcoxon analysis, see Figure 3. In the 2D setting, the difference between SJ and SF in the Dyadic Appropriateness scenario as well as SF and GT in the Naturalness scenario was statistically insignificant. In VR however, only the Dyadic Appropriateness scenario showed statistically insignificant differences, and this was true when comparing GT with SG and SJ with SF.

We can also see from Table 1 that over all systems, over all scenarios, the VR setting scored a slightly higher average rating. Running the Wilcoxon test comparing all 2D ratings with all VR ratings gives a p-value of 0.0068 which indicates that the difference is statistically significant.

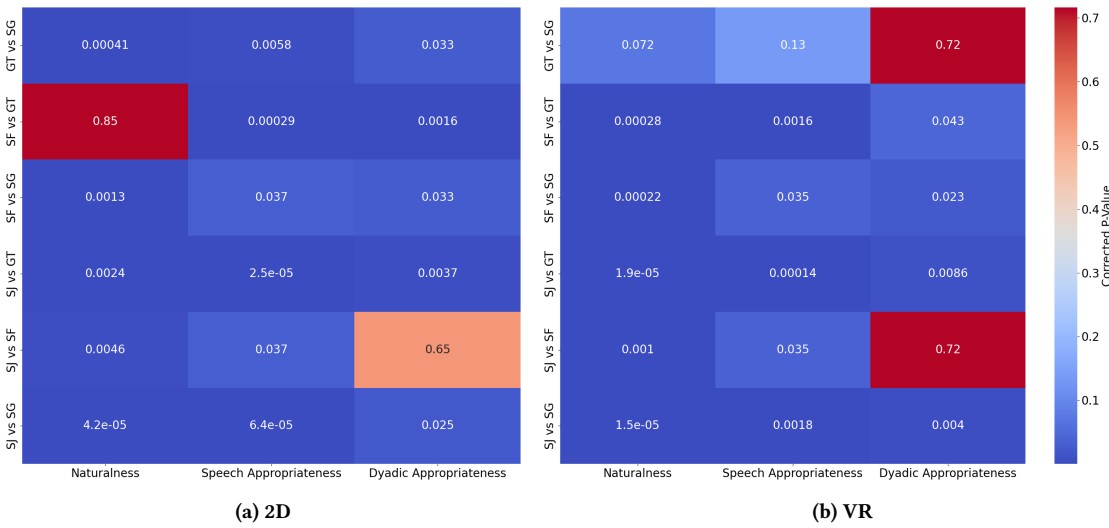

**Figure 3: Wilcoxon comparison results per scenario by setting**

**Table 1: Average Rating Per Setting**

| Setting | Average Rating |
|---------|---------------|
| 2D | 3.11 |
| VR | 3.24 |

## 5.2 Results by system

*5.2.1 System performance in 2D.* Looking at Figure 4a, we can see that systems GT and SG performed best, with average scores of 3.48 and 3.46, respectively, despite SG scoring 55% 4's and 5's while GT only scored 52% 4's and 5's. In contrast, SF exhibited a greater spread with an average score of 2.93, and SJ showed the lowest performance with an average of 2.58. However, a pairwise Wilcoxon comparison of the systems indicates that the difference in rating between GT and SG is not significant (see Table 2). This suggests that the performance of GT and SG is statistically similar, meaning any observed differences are not large enough to be distinguished from random variation in the data. Consequently, no real conclusion can be drawn as to the real ranking of GT and SG based on the measured criteria.

This lack of significant difference implies that evaluators do not perceive one system as superior to the other, and the choice between GT and SG might therefore be based on other factors such as personal preference or whims. The lack of significant difference might also suggest that the observed similarity in performance could be due to random variation, indicating that more sample data would be needed to draw more reliable conclusions.

The other comparisons were all significant, meaning the ranking of the other models (SJ and SF) is otherwise correct, with significant differences observed in their performance compared to GT and SG.

*5.2.2 System performance in VR.* Moving on to the VR performance, it is immediately obvious from Figure 4b that the motion-captured system (GT) outperformed the others with a heavier distribution towards the scores 4 and 5. In fact, 57% of both GT's and SG's scores were 4 or 5 although GT received 8% more 5's. Interestingly, the average score of the GT system was 3.61 while the average of the SG system was 3.62, owing to the comparatively low frequency of 1's and 2's system SG received. The other show very similar performance in VR as in 2D, with only slightly higher average scores (SF: 3.06, SJ: 2.69). Again, the lack of significant difference between GT and SG suggests that both systems perform at a high level, and that their slight differences are not enough to be considered statistically different (see Table 3). However, the significant differences in other pairwise comparisons indicate continued clear performance hierarchies among the other systems.

## 5.3 Results by scenario

Moving on to the rating distribution of the scenarios we can read from Figure 5a that in the 2D setting, the rating distribution is similar between the two dyadic scenarios, while the monadic scenario had a markedly lower distribution of 5's and a markedly higher distribution of 1's and 2's. This might indicate that the actual naturalness of the motions is lacking, compared to the appropriateness

**Table 2: Wilcoxon signed-rank test results for the 2D setting**

| Comparison | P-Value | Corrected P-Value | Significance |
|------------|---------|-------------------|--------------|
| GT vs SG | 8.91e-01 | 8.91e-01 | N |
| GT vs SJ | 2.61e-08 | 1.56e-07 | Y |
| GT vs SF | 8.64e-06 | 3.46e-05 | Y |
| SG vs SJ | 4.66e-08 | 2.33e-07 | Y |
| SG vs SF | 3.43e-05 | 1.03e-04 | Y |
| SJ vs SF | 8.39e-04 | 1.68e-03 | Y |

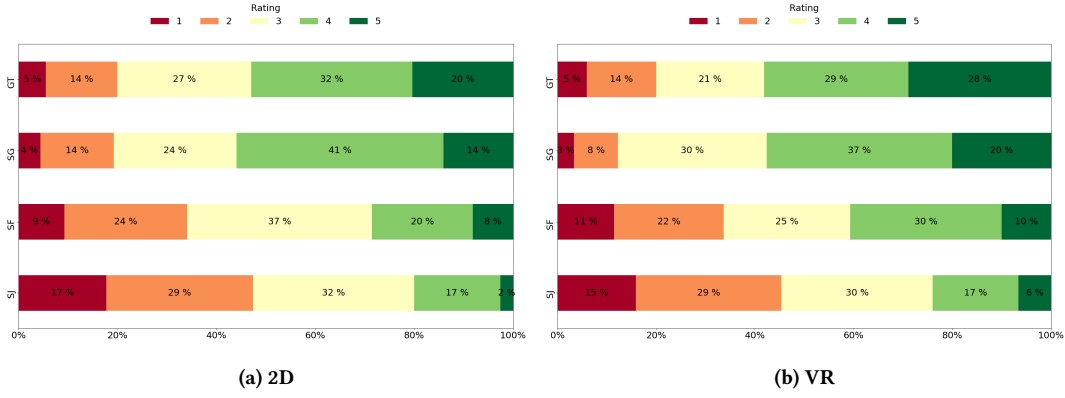

**Figure 4: Distribution of ratings per system and setting**

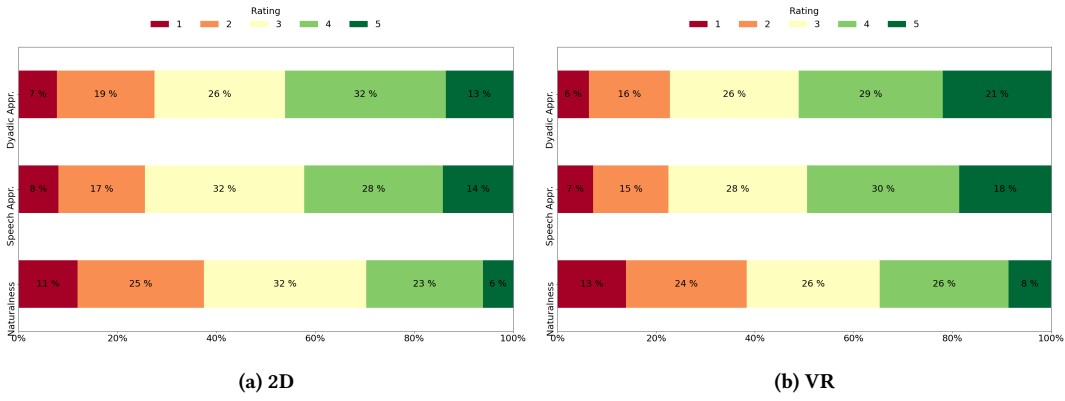

**Figure 5: Distribution of ratings per scenario and setting**

in a conversational setting. It might also indicate that the monadic setup somehow had a negative impact on the subjects perception of the movements. Anecdotally, many participants stated that when viewed from the front it was easier to spot 'clipping' (the avatar moving their arms through other parts of their body, for example) while when viewed from the side, this was less visible. Furthermore, it is obvious that there is significant variability in individual perceptions, indicating differing opinions on the quality and appropriateness of the generated motions in all measured aspects.

The distribution in the VR setting looks a bit different, with a similar spread across all scenarios with the exception of a slightly up-shifted distribution towards the upper scores.

Further comparison shows a slightly higher average rating for each scenario in VR than in 2D (see Table 4), indicating that the setting has some influence on the perception of the generated gestures. However, upon performing a Wilcoxon comparison, the results of which can be found in Table 5, we can see that none of the comparisons cleared the significance threshold which means any difference between the ratings of scenarios in 2D and VR is more likely because of other factors or just random.

**Table 3: Wilcoxon signed-rank test results for the VR setting**

| Comparison | P-Value | Corrected P-Value | Significance |
|---|---|---|---|
| GT vs SG | 8.38e-01 | 8.38e-01 | N |
| GT vs SJ | 7.12e-06 | 3.56e-05 | Y |
| GT vs SF | 1.37e-05 | 5.50e-05 | Y |
| SG vs SJ | 2.79e-06 | 1.67e-05 | Y |
| SG vs SF | 3.27e-05 | 9.81e-05 | Y |
| SJ vs SF | 1.01e-03 | 2.01e-03 | Y |

**Table 4: Average Rating Per Scenario**

| | Average Rating | |
|---|---|---|
| **Scenario** | **2D** | **VR** |
| Naturalness | 2.86 | 2.91 |
| Dyadic Appropriateness | 3.24 | 3.44 |
| Speech Appropriateness | 3.23 | 3.38 |

## 5.4 Inter-rater reliability

The difference in how participants rated systems across the scenarios and ratings is another interesting metric to examine, this time using Kendall's Coefficient of Coherence. To do this, first the data had to be sorted by each participants average rating per system per scenario.

Calculating Kendall's W on this data generated the results in Table 6. A value close to 0 means that there each participant rated very differently, while a value closer to 1 means there was consensus among the participants. In the 2D setting each scenario has a low Inter-rater consensus, meaning the spread of ratings is large for each system. The only scenario which approached consensus was the Naturalness scenario in VR with a W close to 0.6, meaning that most participants gave each system similar ratings. The other two VR scenarios however scored very low, indicating that any given system could score either high or low.

## 5.5 Subjective results

The study also included a post test questionnaire in which the participants, among other things, were asked to give their opinion on the difference between the two settings.

In Figure 6 the results from asking participants whether the setting had positive, negative or no impact on immersion, intelligibility, naturalness and human-likeness are displayed. In Figure 6a we can see that most participants thought the 2D medium had negative or no impact, while in Figure 6b we can see that the participants found an overwhelming positive impact in all categories. Furthermore, 83% of participants preferred VR over 2D. However, this data is

**Table 5: Pairwise Comparison Results of Scenario by Setting**

| Comparison | | | | |
|---|---|---|---|---|
| 2D | VR | P-Value | Corr. P-Value | Sign. |
| Naturalness | Naturalness | 0.750465 | 0.750465 | N |
| Speech Appr. | Speech Appr. | 0.091090 | 0.273271 | N |
| Dyadic Appr. | Dyadic Appr. | 0.102131 | 0.273271 | N |

**Table 6: Inter-rater reliability per scenario and setting**

| Scenario | Kendall's W |
|---|---|
| Naturalness | 0.4026 |
| Speech Appr. | 0.3862 |
| Dyadic Appr. | 0.2058 |
| **(a) 2D** | |

| Scenario | Kendall's W |
|---|---|
| Naturalness | 0.5711 |
| Speech Appr. | 0.2572 |
| Dyadic Appr. | 0.1196 |
| **(b) VR** | |

entirely subjective and the small sample size makes these results less reliable.

## 6 Discussion

### 6.1 The effects of the setting

After visualizing and analyzing the results, it is evident that there are some interesting differences that arise from performing evaluations in a VR environment instead of in 2D. The most important finding in the context of this study is perhaps that the slightly higher average overall rating in the VR setting turned out to be statistically significant. This means that on average, observers found motions viewed in VR to be slightly more accurate than when viewed in 2D, which answers part of the primary RQ.

VR does indeed affect the perception and subsequent rating of computer-generated gestures. The fact that the average rating was slightly higher might indicate that the experience of seeing the avatar in 3D space enhanced the perception of the motions as human-like and appropriate. One of the test participants made a remark after the tests in 2D and VR, stating that in VR, more focus was placed on the avatars' face/upper region than on the rest of the body, while in 2D it was easier to see the full avatar. Although this remark is entirely anecdotal, it does seem to be in line with the research on ECA in VR environments.

The VR setting has been proven to enhance social presence of avatars, which might affect the observer in the way that one participant remarked. The increased presence of the avatar might cause an observer to perceive it as slightly more human, thus altering the way the observer looks at and interprets the avatar.

Furthermore, it is clear from the post-test questionnaire that a majority of the participants preferred the VR setting, which goes some way to answer the second part of the primary RQ, *i.e.,* what benefits either setting could bring 1.1. The VR setting evidently has the benefit of participants preferring it over 2D, which might increase parameters such as engagement and focus. This same benefit might at the same time have caused bias in the ratings, and is a possible explanation why VR scored higher. The VR setting does however come with some clear downsides, such as more time consumption and hardware requirements. The tests had to be performed on location in a lab compared to, for example, the way the evaluation was done for the GENEA Challenge, where the test could be reached online from a home computer. These drawbacks limit the test in terms of scalability, making larger tests much more cumbersome than if done solely in 2D, although increased proliferation of consumer VR/XR devices may change this in the future.

### 6.2 The performance of the models

The general analysis revealed that certain systems consistently performed better than others across all settings and scenarios. Notably, the GT system, which utilized motion-captured data, tended to receive a larger distribution of higher ratings overall, and especially in the VR setting, which has some interesting implications. The GT system was, as previously explained, the 'true movement', and the fact that it scored higher in VR might mean that its human qualities were more evident in VR. This further adds another benefit to VR evaluations in that it might give a more robust ground-truth scenario with which to compare the generated movements. The

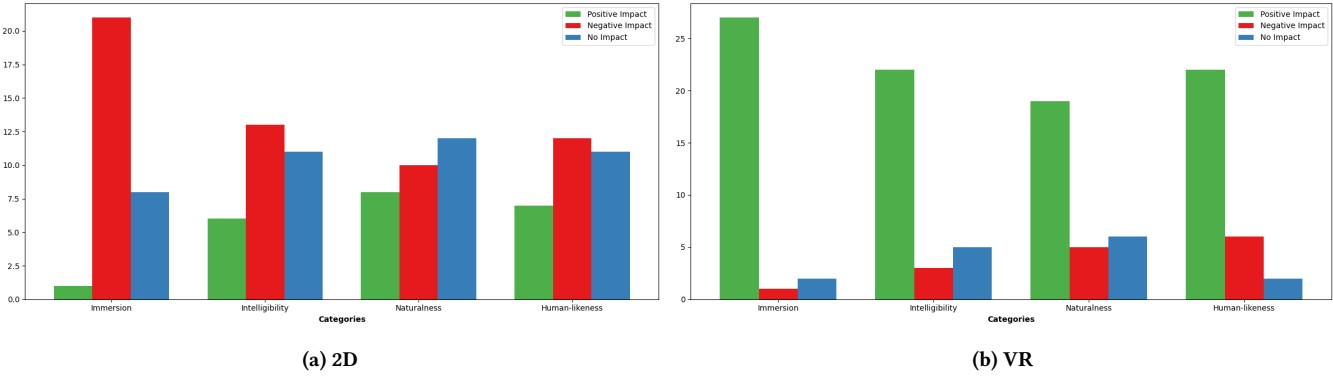

**(a) 2D**
        **(b) VR**

**Figure 6: Distribution of ratings per scenario and setting**

fact that the GT system consistently performed a little above all generation models also indicates that the test participants didn't rate completely by random. It indicates that at least some care and thought went into the rating of the clips, making the results more valid.

Of the generation models, SG consistently performed well, SF consistently performed in the middle, and SJ consistently performed the least well. Furthermore, the Wilcoxon signed-rank test highlighted these differences as significant, meaning that the ranking is correct. This is corroborated by the 2023 GENEA Challenge, in which these models received a similar ranking across most tests [15].

The only insignificant comparisons were between GT and SG. The SG model consistently performed close to the motion-captured clips although GT often got a higher share of 5's than SG did. Given then that this difference was not significant, it is unclear whether the performance of SG in relation to GT is a trustworthy result. Here, there might be outstanding factors, such as the design of the avatars, the lack of sophistication in the animation software, or the performance of the computer that could have equalized the GT movements.

## 7 Conclusions

Although the VR setting seemingly enhanced immersion and caused perception of the generated movements to be more positive, the cumbersome nature of performing such tests compared to 2D is an inherent drawback. Whether the measured statistical difference between the settings is worth that extra trouble is subject to further study. Furthermore, each generation model was measured equally and a statistically significant consistent ranking was established and corroborated, with SG showing the best performance, SF in second place and SJ last. Furthermore, it was established that the true movement system (GT) consistently performed better than all generation models, indicating that participants rated in a reasonable, thought-out manner.

The main limitation of this work is the small sample size. This combined with the fact of the rather large similarities between test subjects (*i.e.,* ethnic affiliation, age, gender, stage in life *etc.*) makes results less reliable. Although the results show some interesting things, it is difficult to say if these conclusions would hold over

a larger demographic. Another possible limitation is the novelty factor of VR. Most of the test participants reported little or no experience in VR. This could have caused a disproportionately positive (or negative) perception of the VR setting as a whole which could potentially have skewed the results. However, it is difficult to state the actual effect or its magnitude.

With the limitations in mind, any future work should seek to recreate or modify this type of comparative evaluation with a much larger sample size. It might also be interesting to include a halfway measure between 2D and VR such as a 3D-viewer where the user can move around in the scene but without full immersion. Furthermore, it could be interesting to examine whether the setting directly influences the subjects engagement, through eye tracking or other measures.

## Acknowledgments

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

Received 20 February 2007; revised 12 March 2009; accepted 5 June 2009