# OpenReview forum: "Gesture Evaluation in Virtual Reality"
_ACM.org/ICMI/2024/Workshop/GENEA — GENEA Workshop 2024_

### Official Review · Reviewer_cy1v · 2024-07-26
**The paper introduces gesture evaluation in VR with some interesting results.**

**Rating:** 6
**Confidence:** 4

**Review:**

### Overview

This paper explores the effect of evaluating gesture generation methods in VR compared to 2D video rendering. Methods are typically ranked higher performing in VR, but significant differences were also found between the two.

### Strengths

This is an interesting evaluation approach and opens further research questions.

The method of evaluation seems sensible and well-considered, with each test justified.

The method appears reproducible, but it would be helpful to publicly release the environment and C# scripts to ensure a consistent test environment.

Testing in VR is novel and could be an exciting topic to explore as the application of ECAs moves towards an interactive virtual setting.

### Questions/Weaknesses

1) The related work mentions fatigue in a past user study. The same would naturally apply to testing in VR. It may be out of scope for this paper, but it would be interesting to know how this fatigue compares. Does fatigue set in faster than in a 2D setting? Was this mentioned in the post participation feedback at all?

2) The Clips chosen for the questions are not the same as those in the GENEA Challenge. This may not be a problem; however, it would be nice if the timings of these clips were released, too.

3) The consensus among participants is surprisingly low. Is there any intuition for why this may be the case? Does Kendall’s W rely on an exact match, i.e. 5 and 5, or does it consider 5 and 4 to be closer in consensus than 5 and 1, for example?

4) It is not clear exactly what the 2D render was. Was this a render of the VR scene, i.e., the same as what the participants saw in VR but in 2D?

5) The clips were presented in a random order, which is good, but were participants shown VR first, then 2D, or vice versa in the same order each time? Would randomising this retain the higher rankings in the VR setting?

6) The quality of the exposition could be improved. There are quite a few typos, particularly regarding the labels of Figures and Tables in the text. The Discussion and Conclusion are similar; Sections 6.2 and 7.2 are exact copies.

L399, 701 figure → Figure

L721 missing .

L790 Furthermore, The → Furthermore, the

L847 its is → it is

### Rating

Overall, the method and approach to answering the research questions seem sound and raise some interesting future considerations. The work is novel and may be helpful in further gesture generation evaluation work. However, as mentioned above, there are a few unanswered questions. These are not significant changes or new experiments required for this paper, so they may be easy to address in the paper. However, the quality of the exposition needs to be improved slightly.

**Nominate For A Reproducibility Award:**

No promise to release code or evalaution environment. The method itself seems reproducible, but the clip timings are not available in the paper, and therefore the experiment would be difficult to exactly reproduce without.

---

### Official Review · Reviewer_CrjX · 2024-07-26
**Interesting attempt to use VR for gesture evaluation, but lacking experimental details**

**Rating:** 5
**Confidence:** 3

**Review:**

### Paper strengths:
- Interesting observation of rating difference between 2D and VR settings.

### Paper weaknesses:
- The reasons why VR scored higher than 2D are not fully convincing to me. The manuscript lacks visual explanations (i.e., figures and images) about the differences between 2D and VR settings. Also, no supplementary materials like videos are provided. Therefore, I could not clearly understand the differences between the two settings. For example, the manuscript explains that "in VR, more focus was placed on the avatars face/upper region ... in 2D it was easier to see the full avatar." Is it because the viewpoint can be moved in VR? If so, would similar results be obtained if a camera position was changed in 2D? In addition, regarding the explanation in lines 719-720, stating "observers found motions viewed in VR to be slightly more accurate than when viewed in 2D," a more detailed explanation is needed to understand why motion is more accurate in VR.
- As the authors also have mentioned, one of the drawbacks of VR is that it is not suitable for large-scale experiments.
- Regarding Sec.3.3, the detailed content of the pilot test is unclear. Was the pilot test conducted for both the 2D and VR settings? If it was only conducted for one of them, it could potentially influence the experimental results.
- Regarding the box plot in Figure 2, all the distributions are integers - I think it’s strange (unintentional rounding?). Also, this box plot does not have median and/or mean values.
- I think several important details about environmental design are missing in the manuscript.
  - The rationale for choosing the office environment is unclear. In other studies on gesture generation, experiments are typically conducted with simple backgrounds to eliminate the influence of the background. This experimental setting does not align with those prior studies.
   - Details about the test environment, such as the VR equipment and the 2D viewing conditions, are not provided.
   - During the experiment, were attention checks conducted for the participants?
   - Was the order of VR and 2D randomized (not only clip order)?
   - While this study uses a 1-5 rating scale, previous studies employed a more detailed 0-100 rating scale. If there is a reason for choosing a different rating scale from prior studies, it should be explained.
- I think Sec.7 (Conclusions) is redundant as it largely repeats the content from Chapters 5 and 6.

### Minor points:
- Genea -> GENEA
- (line 72) ECA’s -> ECAs
- (line 728) avatars face -> avatars’ face

### Additional Comment:
I think that a good environment for gesture evaluation is one that allows for attention to the finer differences in gestures. From this perspective, the evaluations for VR might have been generally lenient than 2D. While this study has demonstrated the potential of VR in gesture evaluation, more detailed discussion and assessment would be needed to determine if VR is a good evaluation environment.

**Nominate For A Reproducibility Award:**

No

---

### Official Review · Reviewer_QR63 · 2024-07-26
**The experiment setup is well-designed, thorough and well-discussed. However, not getting too much insight from the results.**

**Rating:** 6
**Confidence:** 3

**Review:**

Paper Summary
The paper proposed an evaluation approach for synthesized 3D gesture motions using virtual reality. The motivation is to provide a more immersive setup for evaluating gesture motions compared to a traditional 2D setting. The authors investigated the new evaluation setup using the gesture synthesis results from the top three methods from the 2023 GENEA Challenge. The experiment results indicate that gestures observed through VR setup receive higher ratings than 2D. However, the ranking between different methods remain largely identical between both settings and therefore it is not obvious whether the new setup help improving the evaluation results given a higher setup cost.

Strength
The proposed idea for evaluating gestures is well-motivated. Both the experiment designs and implementations are  discussed in details for reproducibility. The paper is well written with both the experiments and analysis being thorough and intuitive to understand. Overall the reviewer finds the paper's experiment interesting and expects this research will motivate discussions in the venue.

Weakness
While the experiment is well-designed and the paper also provides good discussions about the results, overall there is not too much new insight or exciting findings from the experiment results. While it is intuitive to see the VR setup produces a better experience and will be preferred by an user, the result also suggests that it doesn't provide significant improvement in judging the gesture quality from different methods. This is given by the fact of little difference in distribution of ratings between 2D and VR setup. This result indicates that the more laborious VR setup may not be justified since the end evaluation results are not more accurate. Also, there should be a discussion of using desktop 3D viewer as a middle-ground for evaluating gesture motions. This setup will allow similiar3D capability to let the users evaluating 3D gestures from different camera views in real-time, while still scalable for large evaluations through WebGL or similar technologies.

Rating Justification
Overall the paper proposed an interesting evaluation approach based on VR setup. The exposition is clear and both the experiment setup and results are well discussed in details. However, the experiment results do not provide much insight about whether VR setup is superior in evaluating gesture qualities. The paper lacks a comparative discussion with 3D web viewer that has both the simplicity of 2D evaluations and the strength of visualizing 3D gestures from different viewing angles. I still believe the paper will be interesting to the workshop attendee and should inspire more discussions about the ideal setup for gesture evaluations. However, given the above pros and cons, I will be neutral for the paper.

**Nominate For A Reproducibility Award:**

No

---

### Decision · Program_Chairs · 2024-07-30

**Decision:**

Accept

**Comment:**

The paper investigates a novel evaluation approach for synthesized 3D gesture motions using virtual reality (VR), comparing it to traditional 2D settings. By analyzing the top three methods from the 2023 GENEA Challenge, the authors demonstrate that gestures evaluated in VR receive higher ratings than those viewed in 2D. The relative ranking of methods remains unchanged between the two settings. The paper's strengths include its innovative evaluation approach, detailed experiment design, and clear organization, which ensure reproducibility and provide interesting observations that could motivate further research. Nonetheless, the experiment results do not offer substantial new insights or exciting findings, as the ranking between methods remains largely the same in both VR and 2D. Additionally, the manuscript lacks visual explanations and supplementary materials to clearly illustrate differences between the two settings.

I would be in favor of accepting this paper. The authors should, however, correct some of the errors cited by the reviewers. Acceptance is also conditional on modification of the conclusion, which is clearly redundant with the discussion section.